# Antibody Persistence of Human Diploid Cell Rabies Vaccine Administrated Using the Four-Versus Five-Dose Essen Intramuscular Regimen in Post-Exposure Prophylaxis: A Prospective Cohort Study Among the Chinese Population

**DOI:** 10.3390/vaccines13030215

**Published:** 2025-02-21

**Authors:** Linlin Wu, Yu Zhang, Zhuoying Huang, Hongmei Lu, Xiaojun Li, Qi Zhu, Chunli Yin, Jiechen Liu, Huiyong Shao, Xiaodong Sun

**Affiliations:** 1Shanghai Municipal Center for Disease Control & Prevention, Shanghai 200336, China; wulinlin@scdc.sh.cn (L.W.); zhangyu@scdc.sh.cn (Y.Z.); huangzhuoying@scdc.sh.cn (Z.H.); liujiechen@scdc.sh.cn (J.L.); shaohuiyong@scdc.sh.cn (H.S.); 2Shanghai Songjiang District Municipal Center for Disease Control & Prevention, Shanghai 201620, China; sjcdclhm@126.com (H.L.); kindaichi1990@126.com (Q.Z.); 3Shanghai Baoshan District Municipal Center for Disease Control & Prevention, Shanghai 201901, China; jimian@bscdc.org.cn (X.L.); yinchunli@bscdc.org.cn (C.Y.)

**Keywords:** antibody persistence, four-dose Essen regimen, rabies, human diploid cell vaccine

## Abstract

Objective: Evidence on long-term antibody persistence for the rabies vaccine administered using the four-dose Essen regimen is lacking. This study compared antibody persistence for the human diploid cell rabies vaccine (HDCV) administered using the four- versus five-dose Essen intramuscular regimen in post-exposure prophylaxis (PEP). Methods: This prospective cohort study enrolled patients vaccinated with the lyophilized HDCV for PEP who were grouped into four-dose and five-dose Essen groups. Rabies virus-neutralizing antibody (RVNA) detection was performed at 1 year or 3 years after initial vaccination. Results: In total, 180 and 184 patients were included in the four- and five-dose groups, respectively. The 1-year seroconversion (>0.5 IU/mL) rates were similar in the five-dose and four-dose Essen groups (99.2% vs. 98.3%, *p* = 0.662), as were the 3-year seroconversion rates (98.4% vs. 98.3%, *p* > 0.999). The median RVNA titer was significantly higher with the five-dose Essen regimen compared with the four-dose Essen regimen at 1 year (2.75 vs. 4.6 IU/mL, *p* = 0.002), and both groups had similar rates at 3 years (2.00 vs. 3.80 IU/mL, *p* = 0.443). Multivariable stepwise linear regression analysis showed that the five-dose Essen regimen was independently associated with higher serum RVNA titer compared to the four-dose Essen regimen (β = 0.175, *p* = 0.001), and 3 years after vaccination, was independently associated with a lower serum RVNA titer compared to 1 year (β = −1.06, *p* = 0.049). Conclusions: The four- and five-dose Essen regimens effectively produce durable immunogenicity, supporting the feasibility of implementing the four-dose Essen regimen for rabies immunization in China.

## 1. Introduction

Rabies is a zoonotic infectious disease whose mortality rate approximates 100%, making it by far the most fatal acute infectious disease in humans [1,2,3]. Rabies exists in more than 150 countries and territories worldwide, most commonly in Asia and Africa [1], causing almost 59,000 deaths and more than 3.7 million annual losses of disability-adjusted life years (DALYs) [4]. The incubation period of rabies ranges from <1 week to years [1,2,3], and no effective treatment is available for symptomatic rabies [5]. Though exceptional cases where patients survived following clinical rabies have emerged [6,7,8], pre-or post-exposure prevention through vaccination remains the cornerstone of rabies control, especially in endemic regions. Standard rabies vaccination is the only effective measure for rabies post-exposure prophylaxis (PEP) [9]. Currently, the post-exposure rabies vaccine immunization procedures applied in China include the Essen and Zagreb regimens, both recommended in *Technical Guideline for Human Rabies Prevention and Control* (2016 version) [10]. Still, many patients do not complete the immunization due to the multiple doses required and a lack of knowledge about this disease [11,12].

In 2018, following the recommendations of the Advisory Committee on Immunization Practices (ACIP), the World Health Organization (WHO) revised its guidelines for rabies PEP and replaced the traditional five-dose Essen regimen with a simplified “4-dose Essen regimen” for primary exposure to rabies [13]. This adjustment was based on a growing body of evidence suggesting that a four-dose schedule could achieve similar immunogenicity to the five-dose regimen. In a multicenter cross-sectional study conducted in China, patients administered the five-dose Essen regimen exhibited a rabies virus-neutralizing antibody (RVNA) titer above 0.5 IU/mL just 14 days after completing the fourth dose [14]. Furthermore, a previous study demonstrated a 100% seroconversion rate with the four-dose Essen intramuscular regimen, indicating not only the vaccine’s strong immunogenicity but also its favorable safety profile [15]. With fewer doses required, the four-dose regimen is less burdensome for patients, increasing the likelihood that individuals will complete their PEP course without interruption, thereby improving adherence, reducing the logistical burden of vaccination, and enhancing overall public health outcomes. However, while studies assessing the immunogenicity of the simplified four-dose Essen regimen have shown positive results in the short term (7 to 14 days after completion), long-term antibody persistence remains largely undefined [14,15,16]. Moreover, although previous studies have focused on the four-dose Essen regimen primarily in Western populations [15,16], there is a critical need to assess the persistence of antibodies in different demographic groups. Given the distinct epidemiological, environmental, and health system factors in China, as well as potential genetic differences influencing immune response, further investigation into the long-term immunogenicity of the four-dose Essen regimen in Chinese individuals is warranted.

Therefore, the present study aimed to compare antibody titers for the human diploid cell rabies vaccine (HDCV) administered using the four- versus five-dose Essen intramuscular regimen at 1 and 3 years after immunization. The results could provide further evidence for the optimization of immunization strategies and application of the four-dose Essen regimen.

## 2. Methods

### 2.1. Study Design and Participants

This prospective cohort study was conducted from March to December 2023, based on the Shanghai Immunization Planning Information System (SIPIS). To compare antibody titers 1 and 3 years after full immunization between the 5- and 4-dose Essen regimens, a cohort of participants vaccinated during two separate periods (March to December 2020 and March to December 2022) in Songjiang or Baoshan District, Shanghai, were enrolled. Eligible participants must have been vaccinated with the lyophilized HDCV (Chengdu Kanghua Biological Products Co., Ltd. Chengdu, China) after primary exposure. The exclusion criteria were as follows: (1) tumor or autoimmune disease; (2) other rabies vaccination or rabies immunoglobulin detected before or after HDCV vaccination; (3) axillary temperature >37.0 °C at blood sampling. All vaccinees were screened by telephone survey, and eligible participants had to visit the center within 2 weeks for informed consent signing and blood sampling.

The study was approved by the ethics review committee of Shanghai Municipal Center for Disease Control and Prevention (approval #2021-133). All participants provided signed informed consent.

### 2.2. Grouping and Data Collection

Participants were divided into the 5-dose and 4-dose Essen groups. Participants in the 5-dose Essen group were administered five doses on days 0, 3, 7, 14, and 28. Those in the 4-dose Essen group independently chose not to receive the fifth dose (irrespective of reason) and were given four doses on days 0, 3, 7, and 14.

Basic characteristics, including sex, age, date of initial immunization, canine injury, vaccination regimen, passive immunization, family history of allergies, exposure to animals, COVID-19, and influenza vaccination, were collected from the SIPIS and a questionnaire (Figure 1).

### 2.3. Antibody Detection

Blood collection was performed at 1 and 3 years after PEP vaccination, with a one-month time interval from the original vaccination. Venous blood (5 mL) was collected from the participants, and the serum was obtained by centrifugation at 3000 rpm for 5 min after 30 min of clotting. The serum was stored at −20 °C. The serum samples were tested in batches using the WHO-recommended rapid fluorescent-focus inhibition test (RFFIT) method [17], with a detection limit of 0.1 IU/mL. One positive control and one negative control were set for each test plate. According to the WHO standard, RVNA ≥ 0.5 IU/mL was considered to indicate seroconversion [7]. The test was performed by Wuhan Biological Products Research Institute Co., Wuhan, China.

### 2.4. Statistical Analysis

Categorical variables were described as *n* (%) and compared using the chi-square test or Fisher’s exact test. The Shapiro–Wilk method was used to assess the normality of continuous data. Continuous variables with a normal distribution are presented as the mean ± standard deviation (SD) and were analyzed using the independent samples *t*-test or analysis of variance (ANOVA); those with a skewed distribution are presented as the median (interquartile range [IQR]) and were analyzed using the Mann–Whitney U-test or the Kruskal–Wallis H-test. The Bonferroni method was used for pairwise comparisons. Linear regression analysis was performed with log(IgG) as the dependent variable. The multivariable analysis included variables with univariable *p* < 0.10. For antibody concentrations below the lower detection limit, imputation with 0.05 IU/kg (half of the lower detection limit) was performed. SPSS 22.0 (IBM, Armonk, NY, USA) was used for data analysis, with two-sided *p* < 0.05 considered statistically significant.

## 3. Results

### 3.1. Characteristics of the Participants

A total of 364 patients were enrolled in this study, including 180 in the four-dose group and 184 in the five-dose group (Figure 1). The median ages were 33.5 years (IQR, 18–51) and 41 years (IQR, 12–63) in the four-dose and five-dose Essen groups, respectively. The proportions of children (<14 years) and elderly people (≥60 years) were higher in the five-dose Essen group compared with the four-dose Essen group (*p* < 0.001), as was the proportion of influenza vaccination (*p* = 0.005). A higher proportion of family pet bites was reported in the four-dose Essen group than that in the five-dose Essen group (*p* = 0.049). There were no significant differences between the two groups in terms of sex, body mass index, time of vaccination, allergic diseases, and family history of allergies (all *p* > 0.05) (Table 1).

### 3.2. Seroconversion

Serum RVNA ≥0.5 IU/mL was found in 359 out of 364 patients, indicating a seroconversion rate of 98.6%. There was no significant difference in seroconversion rate between the four-dose and five-dose Essen groups (98.3% vs. 98.9%, *p* = 0.682) (Table 2). Considering the time from vaccination, the 1-year seroconversion rates were similar between the five-dose and four-dose Essen groups (99.2% vs. 98.3%, *p* = 0.622), as were the 3-year seroconversion rates (98.4% vs. 98.3%, *p* > 0.999) (Table 2).

### 3.3. Antibody Titer

Serum RVNA concentrations in the 364 patients ranged from below the detection limit to 128.3 IU/mL, with a median concentration of 3.70 (IQR, 1.60–6.03) IU/mL. The median concentration was 2.20 (IQR, 1.50–5.00) IU/mL in the four-dose Essen group, versus 4.40 (IQR, 1.90–8.30) IU/mL in the five-dose Essen group (*p* = 0.001) (Table 2).

At the 1- and 3-year time points, median serum RVNA titers in the four-dose Essen group were 2.75 (IQR, 1.58–5.60) IU/mL and 2.00 (IQR, 1.50–4.40) IU/mL, respectively, with no significant difference observed (*p* > 0.999). Similarly, there was no significant difference in serum RVNA concentrations between the 1- and 3-year time points in the five-dose Essen group, i.e., 4.60 (IQR, 2.60–8.90) IU/mL versus 3.80 (IQR, 1.70–6.80) IU/mL, respectively (*p* = 0.172). The median antibody titer was significantly higher in the five-dose Essen group compared with the four-dose Essen group (2.75 vs. 4.60 IU/mL, *p* = 0.002) at 1 year, but no significant difference was observed between the four-dose and five-dose Essen groups at 3 years (2.00 vs. 3.80 IU/mL, *p* = 0.443) (Table 2 and Figure 2).

The serum RVNA titers were log-transformed to perform univariable and multivariable linear analyses (Table 3). Multivariable stepwise linear regression analysis showed that the five-dose Essen regimen was independently associated with a higher serum RVNA titer compared to the four-dose Essen regimen (β = 0.175, *p* = 0.001), and the time elapsed from vaccination at 3 years was independently associated with a lower serum RVNA titer compared to 1 year (β = −1.06, *p* = 0.049).

## 4. Discussion

The development of human rabies vaccines has been rapid in recent years, from early neural tissue vaccines to embryonated egg-based rabies vaccines (CCEEVs) and cell culture, which have higher immunogenicity and improved safety while simplifying immunization procedures. Historically, rabies PEP involved a series of 14–21 doses, which proved to be cumbersome, especially in resource-limited settings. In response, the number of doses required for effective PEP has been progressively reduced, beginning with the introduction of the five-dose “Essen regimen” (1-1-1-1-1) [13,18,19,20], and further simplified to the “four-dose Essen regimen” (1-1-1-1) [13,18,19,20]. Compared to the five-dose regimen, the four-dose Essen regimen has demonstrated superior adherence rates. In a study by Shankaraiah et al. [21], fewer participants completed the fifth dose of the Essen regimen than the fourth dose (60% vs. 69.8%). This difference highlights a key advantage of the four-dose regimen: it reduces the number of visits required for vaccination, thus improving patient compliance. The reduced number of doses also alleviates the financial burden of frequent hospital visits, including transportation costs, which are particularly significant in rural or underserved areas. In fact, the analysis of factors affecting adherence in this study suggested that the four-dose regimen might be a more practical and sustainable alternative to the five-dose regimen, particularly in regions where access to healthcare may be limited or challenging. Moreover, the four-dose Essen regimen has been shown to be more cost-effective than the five-dose regimen, an important consideration for countries and healthcare systems that face financial constraints [21]. According to the ACIP, assuming 100% adherence to the vaccination protocol, the use of the four-dose Essen regimen instead of its five-dose counterpart could lead to annual cost savings of approximately USD 16.6 million in the United States alone [22]. This potential for cost savings, combined with the improved patient adherence and simplified vaccination schedule, makes the four-dose regimen an attractive option for public health programs aiming to reduce the burden of rabies worldwide, especially in low-resource settings.

As an observational study, the choice of the four-dose or five-dose Essen regimen was determined by participants, since we included participants who did not to receive the fifth dose irrespective of reason in the four-dose group. Therefore, the disparity of baseline characteristics between the two groups somehow reflects the features of the patients. For instance, the patients who completed the five-dose Essen regimen might have better adherence, and be more attentive to their own health. This was supported by the higher proportion of participants with flu vaccination in the five-dose Essen group than that in the four-dose group (83.7% vs. 75.0%). Noticeably, the proportion of children and older adults was significantly higher in the five-dose Essen group compared to the four-dose Essen group (32.6% vs. 21.7% and 34.8% vs. 18.9%, respectively), suggesting that age might be associated with adherence. Whether the patients’ compliance and attention to their health can influence their medical behavior, and subsequently impact long-term prognosis, still requires further investigation.

Regarding the immunogenicity of the four-dose Essen regimen, previous studies have mainly focused on short-term immunogenicity (7 to 180 days) [14,15,16,23,24]. In a cross-sectional study in China, Song et al. [24] measured serum RVNA concentrations 14 days after the fourth dose and 14 days after the fifth dose during the five-dose Essen regimen, reporting a seroconversion rate of 100% at both time points. Haradanahalli et al. [15] examined the immunogenicity of the four-dose Essen regimen but only measured serum RVNA concentrations at 14, 90, and 180 days after PEP, and found that all cases in both the four- and five-dose Essen groups seroconverted from 14 to 180 days after immunization, without a significant difference in the geometric mean concentration (GMC) of RVNA between the two groups. The present study fills the gap of long-term antibody persistence studies of the four-dose Essen regimen. By comparing immunogenicity between the four- vs. five-dose Essen regimens at 1 and 3 years after PEP, the present study detected seroconversion rates >98% in both groups at 1 and 3 years, with no significant differences between them. These findings may be related to the better immune persistence of HDCV [25,26,27,28], as supported by Vodopija et al. [28]. Indeed, serum RVNA concentrations for HDCV were higher than those of the primary chick embryo cell vaccine (PCECV), purified duck-embryo vaccine (PDEV), and purified vero-cell rabies vaccine (PVRV) at 35 and 1100 days after immunization, as well as at 14 days after a booster dose (1114 days) [28]. Zhang et al. [27] found that after 1, 3, and 5 years of PVRV immunization, the seroconversion rates were 90.5%, 49.1%, and 34%, respectively. Meanwhile, Hu et al. [25] reported a seroconversion rate of 98.3% 10 years after HDCV immunization. Overall, vaccination with HDCV, which demonstrates better immune persistence, using both the five-dose and four-dose Essen regimens, can provide durable immunogenicity. Additionally, the four-dose Essen regimen offers a more convenient procedure.

In the present study, the median serum RVNA titer was slightly lower for the four-dose Essen regimen compared with the five-dose Essen regimen at 1 year after PEP, but there was no significant difference between the two groups at 3 years. This may be due to an inconsistent decrease in serum RVNA concentration over time. In agreement, Zhang et al. [27] found that the levels of the rabies antibody geometric mean titer (GMT) at 7 days, 14 days, 45 days, 1 year, 2 years, 3 years, 4 years, and 5 years after immunization with the Essen regimen were 0.56, 8.87, 16.13, 1.79, 1.44, 1.21, and 0.81 IU/mL, respectively. Among these, the GMT levels decreased dramatically from 16.13 IU/mL to 1.79 IU/mL between 45 days and 1 year. Other studies of the five-dose Essen regimen also suggested that the GMC or GMT of serum RVNA peaked between 14 and 90 days after immunization and then declined dramatically over the next year [15,29]. Meanwhile, the decline may occur slightly earlier in the four-dose Essen regimen compared with the five-dose regimen, resulting in serum RVNA titers 1 year after vaccination being slightly lower for the four-dose Essen regimen compared with the five-dose regimen group in the present study. In addition, Fayaz et al. [30] found that 32 years after immunization with HDCV using the five-dose Essen regimen, serum RVNA titers in nine patients who had never received booster injections ranged from 0.3 to 0.98 IU/mL. Suwansrinon et al. [31] reported serum RVNA GMT >0.5 IU/mL in patients administered the three-dose pre-exposure prophylaxis (PrEP) and the five-dose PEP in a study of the long-term immunogenicity of the rabies vaccine over a 5- to 21-year period. Hu et al. [25] found that 10 years after immunization with the five-dose Essen regimen, the GMC of RVNA ranged from 2.18 to 2.44 IU/mL. Overall, it can be hypothesized that after full immunization, the decline in serum RVNA titers in vaccinees would level off rapidly within 1 year and then converge to a threshold value that would allow for mounting a rapid immune response to subsequent booster doses. This might explain why there was no significant difference in serum RVNA titers 3 years after vaccination between the four- and five-dose Essen groups in the present study; however, the current study also indicates numerically higher RVNA titers with the five-dose regimen than the four-dose regimen, which should be further verified in future studies.

Considering the factors that influence immunogenicity, several studies have explored the relationship between various demographic and clinical characteristics and the immune response to rabies vaccination. One such study, which included patients with third-level exposure to rabies virus who received a combined regimen of rabies vaccines and rabies immunoglobulin, found no significant differences in serum RVNA GMT between obese and normal-weight individuals at 7, 14, and 28 days post-vaccination [32]. This suggests that, for patients receiving rabies PEP, factors such as body weight may not significantly affect the immune response. Similarly, in the present study, we did not find any association between the weight or sex of vaccinated individuals and serum RVNA titers. In contrast, age has been more consistently shown to affect the immune response to vaccines. Previous studies have demonstrated that older adults typically exhibit a reduced immune response due to age-related immunosenescence, which leads to lower antibody titers and diminished seroconversion rates following vaccination [33,34]. In the context of the rabies vaccine, Banovic et al. [16] specifically assessed the four-dose Essen regimen and identified age as a key factor influencing serum RVNA titers. Their study found that for each additional year of age, the serum RVNA seroconversion rate decreased by 3.18%, indicating that older individuals may have a somewhat diminished response to the rabies vaccine, which echoes the findings from previous studies that older adults may require higher or more frequent doses of vaccines to achieve the same level of protection as younger individuals [35]. Though the age distribution disparity between the two groups might introduce bias when directly comparing the immunogenicity of the five-dose and four-dose Essen regimens, we did not observe a similar association between age and RVNA titer. The different observations between studies could be attributed to several factors, including differences in study populations, sample sizes, and age distributions between our study and the one by Banovic et al. [16]. For example, the mean ages in the present study were 33 years and 41 years for the four-dose and five-dose groups, respectively, which may be lower compared to the age range in the Banovic et al. study, where older age groups might have been more prevalent [16]. Since older adults and children exhibited robust immune responses to the rabies vaccine in both groups, the four-dose Essen regimen may also offer effective protection across a wide range of age groups. Furthermore, we also assessed whether other factors such as allergic diseases or influenza vaccination history influenced the immune response. Consistent with findings from previous research [36], our study showed no significant effect of these variables on the production of serum neutralizing antibodies. This suggests that the immunogenicity of the rabies vaccine is generally robust and universal, unaffected by common comorbidities like allergies or past influenza vaccination. Instead, the number of vaccine doses and the timing between doses appear to be the primary factors influencing the immune response.

While both pre-exposure and post-exposure rabies vaccination are highly effective in inducing long-term immunity, re-vaccination following re-exposure is still needed in certain circumstances. According to the WHO guidelines, individuals who have received a complete rabies vaccination series (either pre-or post-exposure) typically require two booster doses (usually one site intramuscularly on days 0 and 3) if repeat exposure occurs >3 months after the last vaccination [13]. This recommendation is based on the anamnestic immune response, where the immune system rapidly mounts a strong defense upon subsequent exposure. In a previous study, antibody titers after intradermal rabies vaccination declined over time, with seroprotection rates decreasing as much as 44% after 1.8–2.5 years. Interestingly, a booster dose after this period restores 100% seroprotection [37]. However, in certain situations (such as when the time since the last vaccination is prolonged, the individual’s immune status is uncertain, or the initial immunization is incomplete), a complete immunization procedure after re-exposure may still be strongly recommended. This approach ensures that individuals are sufficiently protected in the event of re-exposure to the rabies virus. In the present study, no participant had subsequent exposure to rabies after initial immunization.

The present study had several limitations. First, serum antibody titers before PEP vaccination could not be examined. Nevertheless, only rabies-vaccinated patients were included in this work, and a meta-analysis examining 80 studies found that essentially all immunologically naïve patients had pre-immunization serum RVNA concentrations below 0.5 IU/mL [35]. Furthermore, each individual only had one serum neutralizing antibody test at 1 or 3 years after vaccination, with a lack of data on short-term antibody titer and the dynamic change in serum RVNA over time. Finally, although previous studies have suggested that the four-dose Essen regimen has a favorable safety profile, safety data were not available in our study as we started after the completion of immunization, which limits our ability to comment on the safety aspects in this population. Additionally, information on sources of exposure was not collected, and there were suggestive differences in the characteristics of the two groups (e.g., age distribution and influenza vaccination adherence) that may have introduced confounding biases. While these factors were adjusted for using various statistical methods, potential residual confounding cannot be ruled out.

In conclusion, the four- and five-dose Essen regimens effectively produce a durable immunogenicity, with all seroconversion rates > 98% at 1 and 3 years after PEP, supporting the feasibility of the four-dose Essen regimen for rabies immunization in China. Prospective studies with repeated measurements should be conducted to determine the dynamic changes in serum antibody titer and clarify the long-term immunogenicity of the four-dose Essen regimen.

## Figures and Tables

**Figure 1 vaccines-13-00215-f001:**
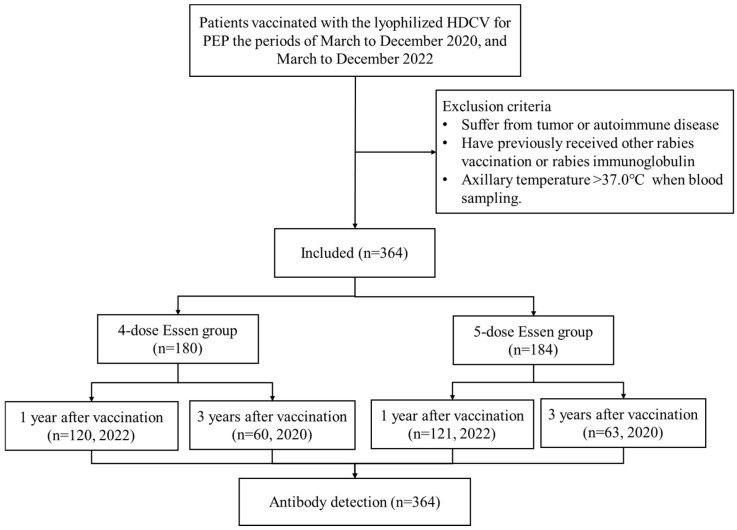
Study flowchart. HDCV: human diploid cell rabies vaccine; PEP: post-exposure prophylaxis.

**Figure 2 vaccines-13-00215-f002:**
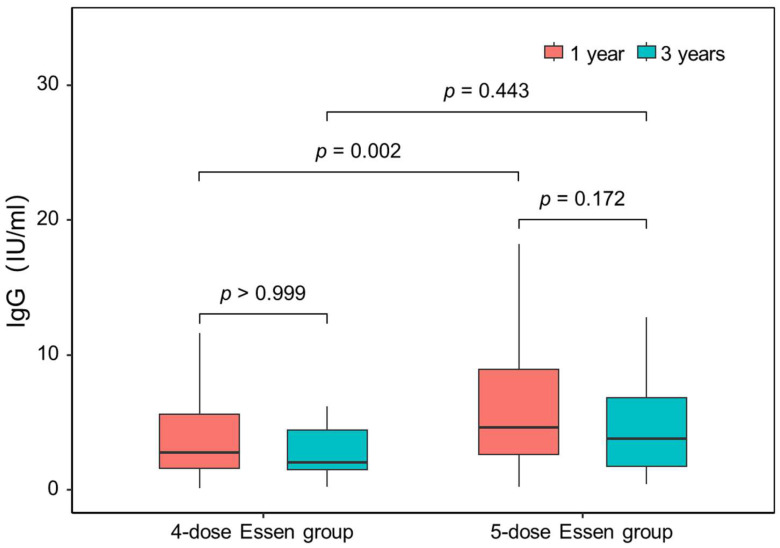
Logarithmic rabies virus-neutralizing activity (RVNA) titers 1 and 3 years after vaccination in the 4- and 5-dose Essen groups.

**Table 1 vaccines-13-00215-t001:** Characteristics of the patients. BMI: body mass index.

Variable	Four-Dose Essen (*n* = 180)	Five-Dose Essen (*n* = 184)	*p*
Age, years, *n* (%)			<0.001
<14	39 (21.7)	60 (32.6)	
14–59	107 (59.4)	60 (32.6)	
≥60	34 (18.9)	64 (34.8)	
Sex, *n* (%)			0.619
Male	78 (43.3)	75 (40.8)	
Female	102 (56.7)	109 (59.2)	
BMI, kg/m^2^, *n* (%)			0.194
<18.5	29 (16.1)	38 (20.7)	
≥18.5, <24	92 (51.1)	77 (41.8)	
≥24	59 (32.8)	69 (37.5)	
Time from vaccination, *n* (%)			0.855
1 year	120 (66.7)	121 (65.8)	
3 years	60 (33.3)	63 (34.2)	
Allergies, *n* (%)			0.423
Yes	146 (81.1)	143 (77.7)	
No	34 (18.9)	41 (22.3)	
Family allergies, *n* (%)			0.313
Yes	143 (79.4)	155 (84.2)	
No	16 (8.9)	16 (8.7)	
Unknown	21 (11.7)	13 (7.1)	
Flu vaccination, *n* (%)			0.005
Yes	135 (75.0)	154 (83.7)	
No	33 (18.3)	29 (15.8)	
Unknown	12 (6.7)	1 (0.5)	
Family pet bites, *n* (%)			0.049
Yes	72 (40.0)	55 (29.9)	
No	76 (42.2)	101 (54.9)	
No pet	32 (17.8)	28 (15.2)	

**Table 2 vaccines-13-00215-t002:** Antibody titers.

	Four-Dose Essen			Five-Dose Essen			P_1_	P_2_	P_3_
	One Year (*n* = 120)	Three Years (*n* = 60)	Total (*n* = 180)	One Year (*n* = 121)	Three Years (*n* = 63)	Total (*n* = 184)			
Seroconversion, *n* (%)							0.682	0.622	>0.999
Negative	2 (1.7)	1 (3.2)	3 (1.7%)	1 (0.8)	1 (1.6)	2 (1.1%)			
Positive	118 (98.3)	59 (98.3)	177 (98.3%)	120 (99.2)	62 (98.4)	182 (98.9%)			
RVNA, IU/mL, median (IQR)	2.75 (1.58, 5.60)	2.00 (1.50, 4.40)	2.20 (1.50, 5.00)	4.60 (2.60, 8.90)	3.80 (1.70, 6.80)	4.40 (1.90, 8.30)	0.001	0.002	0.443

RVNA: rabies virus-neutralizing antibody; IQR: interquartile range. P_1_: 4-dose Essen total vs. 5-dose Essen total. P_2_: 4-dose Essen at 1 year vs. 5-dose Essen at 1 year, using the Bonferroni correction method. P_3_: 4-dose Essen at 3 years vs. 5-dose Essen at 3 years, using the Bonferroni correction method.

**Table 3 vaccines-13-00215-t003:** Univariable and multivariable linear regression analyses of log (IgG).

Variables	Univariable Analysis			Multivariable Analysis		
	β	Standard Error	*p*	β	Standard Error	*p*
Age						
0–13	−0.009	0.063	0.884			
14–59	reference					
60 and above	0.011	0.063	0.858			
BMI						
<18	0.004	0.071	0.960			
[18.5–24)	reference					
≥24	−0.015	0.058	0.797			
Sex						
Female	reference					
Male	−0.081	0.052	0.122			
Procedure						
4-dose	reference					
5-dose	0.174	0.051	0.001	0.175	0.051	0.001
Time from vaccination						
1 year	reference			reference		
3 years	−0.104	0.054	0.057	−0.106	0.054	0.049
Allergies						
No	reference					
Yes	0.063	0.064	0.323			
Family allergies						
No	reference					
Yes	0.080	0.093	0.387			
Flu vaccination						
No	reference					
Yes	0.059	0.069	0.396			
Family pet bite						
No	reference					
Yes	0.077	0.054	0.157			

BMI: body mass index.

## Data Availability

The original contributions presented in this study are included in the article. Further inquiries can be directed to the corresponding author.

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
