# Peer review of "Antibody Persistence of Human Diploid Cell Rabies Vaccine Administrated Using the Four-Versus Five-Dose Essen Intramuscular Regimen in Post-Exposure Prophylaxis: A Prospective Cohort Study Among the Chinese Population"

_vaccines, 2025, doi:10.3390/vaccines13030215_

Round 1

Reviewer 1 Report

Comments and Suggestions for Authors

I read with great interest the manuscript submitted to me for review given the particularly stimulating topic.

The research is scientifically valid and well-conceived. The conclusions are sound with the experimental design.

However, there remains a difficult evidence to demonstrate, namely, is the antibody titer induced by the vaccine (pre or post exposure) sufficient to protect the subject from the bite of a rabid animal (if pre) or from a subsequent bite if post?

In any case, given the validity of the research, I have some observations to make:

Abstract: nothing to report.

Introduction: it would be worth remembering that some (lucky) cases of recovery with manifest disease are reported in the literature (1. Recovery of a patient from clinical rabies--Wisconsin, 2004. Centers for Disease Control and Prevention (CDC).MMWR Morb Mortal Wkly Rep. 2004 Dec 24;53(50):1171-3.

2. Survival after treatment of rabies with induction of coma. Willoughby RE Jr, Tieves KS, Hoffman GM, Ghanayem NS, Amlie-Lefond CM, Schwabe MJ, Chusid MJ, Rupprecht CE.N Engl J Med. 2005 Jun 16;352(24):2508-14. doi: 10.1056/NEJMoa050382.

3. Recovery of a patient from clinical rabies--California, 2011. Centers for Disease Control and Prevention (CDC).MMWR Morb Mortal Wkly Rep. 2012 Feb 3;61(4):61-5.)

There is also pre-exposure vaccination, but in any case post-exposure prophylaxis is necessary. The vaccine (Long-term serological response and boostability of intradermal rabies immunization: A retrospective chart review.

Burnett EJ, Mcpherson RJ, Aquin JP, Xu KY, Plourde PJ.Can J Public Health. 2024 Nov 26. doi: 10.17269/s41997-024-00968-5. ) certainly induces a high antibody titer, but is it sufficient to protect from the bite of a rabid animal without post-exposure prophylaxis?

Mat & Meth: Are the sources of exposure known? (see also below)

Results: Table 1 indicates per bite yes/no/no pet. Is it known from which animals?

Discussion: Good discussion of results. Acceptable weaknesses. Conclusions are sound with experimental design.

Author Response

Reviewer #1

I read with great interest the manuscript submitted to me for review given the particularly stimulating topic.
The research is scientifically valid and well-conceived. The conclusions are sound with the experimental design.
However, there remains a difficult evidence to demonstrate, namely, is the antibody titer induced by the vaccine (pre or post exposure) sufficient to protect the subject from the bite of a rabid animal (if pre) or from a subsequent bite if post?
    Response: We sincerely thank the Reviewer for their thoughtful feedback and for highlighting this important point. Our study focuses on the post-exposure prophylaxis (PEP) scenario, and we agree that demonstrating protection against a real rabid animal bite is challenging. However, we followed the recommended threshold for rabies virus-neutralizing antibodies (RVNA ≥ 0.5 IU/mL), which is widely recognized as indicative of protective immunity by guidelines worldwide [1-3]. This standard has been validated in numerous studies as sufficient to prevent rabies onset following exposure. We hope this explanation addresses your concern, and we remain open to further suggestions.

In any case, given the validity of the research, I have some observations to make:
Abstract: nothing to report.
Introduction: it would be worth remembering that some (lucky) cases of recovery with manifest disease are reported in the literature (1. Recovery of a patient from clinical rabies--Wisconsin, 2004. Centers for Disease Control and Prevention (CDC).MMWR Morb Mortal Wkly Rep. 2004 Dec 24;53(50):1171-3.
2. Survival after treatment of rabies with induction of coma. Willoughby RE Jr, Tieves KS, Hoffman GM, Ghanayem NS, Amlie-Lefond CM, Schwabe MJ, Chusid MJ, Rupprecht CE.N Engl J Med. 2005 Jun 16;352(24):2508-14. doi: 10.1056/NEJMoa050382.
3. Recovery of a patient from clinical rabies--California, 2011. Centers for Disease Control and Prevention (CDC).MMWR Morb Mortal Wkly Rep. 2012 Feb 3;61(4):61-5.)
    Response: We sincerely thank the Reviewer for their valuable suggestion. As recommended, we have revised the introduction to include information on rare cases of survival following clinical rabies, emphasizing the exceptional nature of these instances. The revised text reinforces the critical importance of prevention through vaccination as the primary strategy for rabies control.

Revised introduction

Rabies is a zoonotic infectious disease, whose mortality rate approximates 100%, making it by far the most fatal acute infectious disease in humans [4-6]. Rabies exists in more than 150 countries and territories worldwide, most commonly in Asia and Africa [4], causing almost 59,000 deaths and more than 3.7 million annual loss of disability-adjusted life years (DALYs) [7]. The incubation period of rabies ranges from <1 week to years [4-6], and no effective treatment is available for symptomatic rabies [8]. Though exceptional cases where patients survived following clinical rabies have emerged [9-11], pre- or post-exposure prevention through vaccination remains the cornerstone of rabies control, especially in endemic regions.

There is also pre-exposure vaccination, but in any case postexposure prophylaxis is necessary. 

The vaccine (Long-term serological response and boostability of intradermal rabies immunization: A retrospective chart review.Burnett EJ, Mcpherson RJ, Aquin JP, Xu KY, Plourde PJ.Can J Public Health. 2024 Nov 26. doi: 10.17269/s41997-024-00968-)certainly induces a high antibody titer, but is it sufficient to protect from the bite of a rabid animal without post-exposure prophylaxis?

Response: We sincerely thank the Reviewer for their thoughtful suggestion. Indeed, our study specifically focuses on PEP, and we agree that while pre-exposure vaccination can certainly induce a high antibody titer, its ability to protect from a rabid animal bite without subsequent PEP remains an important question. According to the World Health Organization (WHO) guidelines, individuals who have previously been received a complete vaccination series (either pre-exposure or post-exposure) typically do not need the full course of rabies vaccination upon subsequent exposure. Instead, they are recommended to receive two booster doses (usually 1-site intramuscularly on day 0 and 3) if repeat exposure occurring>3 months after last vaccination. The rationale is that an anamnestic response will occur, rapidly boosting the immune response in these individuals. This approach ensures that the individual’s immune system is adequately prepared to respond in case of exposure to the rabies virus.

       In our study, we followed the recommended threshold for rabies virus-neutralizing antibodies (RVNA ≥ 0.5 IU/mL) and no participant had subsequent re-exposure to potential rabies virus after PEP. Since the booster dose administration is not within the scope of this study, we have incorporated the reference you mentioned into the Discussion of the revised manuscript to further enhance the context of rabies vaccination strategies.

Revised Discussion:

While both pre-exposure and post-exposure rabies vaccination is highly effective in inducing long-term immunity, re-vaccination following re-exposure is still needed in certain circumstances. According to the WHO guidelines, individuals who have received a complete rabies vaccination series (either pre- or post-exposure) typically require two booster doses (usually 1-site intramuscularly on day 0 and 3) if repeat exposure occurring >3 months after last vaccination [1]. This recommendation is based on the anamnestic immune response, where the immune system rapidly mounts a strong defense upon subsequent exposure. In a previous study, antibody titers after intradermal rabies vaccination declined over time, with seroprotection rates decreasing as much as 44% after 1.8-2.5 years. Interestingly, a booster dose after this period restores 100% seroprotection [12]. However, in certain situations (such as when the time since the last vaccination is prolonged, the individual's immune status is uncertain, or the initial immunization is incomplete), a complete immunization procedure after re-exposure may still be strongly recommended. This approach ensures that individuals are sufficiently protected in the event of re-exposure to the rabies virus. In the present study, no participant had subsequent exposure to rabies after initial immunization.

Mat & Meth: Are the sources of exposure known? (see also below)

         Response: We sincerely thank the Reviewer for this fair comment. We agree that the sources of exposure could be informative. Nevertheless, we only collected data on whether the patient was bitten by a pet and did not gather information on the species of the animal causing the injury. This was mainly because all patients included in our study were from the urban area of Shanghai, where wildlife injury incidents are rare. This was properly discussed in the limitation (see also below).

Results: Table 1 indicates per bite yes/no/no pet. Is it known from which animals?

         Response: We sincerely thank the Reviewer. As our response to the last comment, we only collected data on whether the patient was bitten by a pet and did not gather information on the species of the animal causing the injury. This was mainly because all patients included in our study were from the urban area of Shanghai, where wildlife injury incidents are rare. This was properly discussed in the limitation (see also below).

Revised Discussion:

Additionally, the information on sources of exposure was not collected and there were suggestive differences in the characteristics of the two groups (e.g., age distribution and influenza vaccination adherence) that may have introduced confounding biases. While these factors were adjusted for using various statistical methods, potential residual con-founding cannot be ruled out.

Discussion: Good discussion of results. Acceptable weaknesses. Conclusions are sound with experimental design.

       Response: We sincerely thank the Reviewer for your positive feedback. We are glad that the discussion of the results and the conclusions drawn from the experimental design were appreciated. We have noted the mention of weaknesses and will continue to ensure that future work further addresses these aspects.

References

1       World Health Organization. Rabies vaccines: WHO position paper, April 2018 - Recommendations. Vaccine. 2018, 36, 5500-3.

2       Zhou, H.; Li, Y.; Chen, R.F.; Tao, X.Y.; Yu, P.C.; Cao, S.C.; Li, L.; Chen, Z.H.; Zhu, W.Y.; Yin, W.W., et al. [Technical guideline for human rabies prevention and control (2016)]. Zhonghua Liu Xing Bing Xue Za Zhi. 2016, 37, 139-63.

3       Rupprecht, C.E.; Gibbons, R.V. Clinical practice. Prophylaxis against rabies. N Engl J Med. 2004, 351, 2626-35.

4       Jackson, A.C. Human Rabies: a 2016 Update. Curr Infect Dis Rep. 2016, 18, 38.

5       Hemachudha, T.; Ugolini, G.; Wacharapluesadee, S.; Sungkarat, W.; Shuangshoti, S.; Laothamatas, J. Human rabies: neuropathogenesis, diagnosis, and management. Lancet Neurol. 2013, 12, 498-513.

6       Crowcroft, N.S.; Thampi, N. The prevention and management of rabies. BMJ. 2015, 350, g7827.

7       Hampson, K.; Coudeville, L.; Lembo, T.; Sambo, M.; Kieffer, A.; Attlan, M.; Barrat, J.; Blanton, J.D.; Briggs, D.J.; Cleaveland, S., et al. Estimating the global burden of endemic canine rabies. PLoS Negl Trop Dis. 2015, 9, e0003709.

8       Centers for Disease Control and Prevention. Rabies. https://www.cdc.gov/rabies/index.html. Accessed August 16, 2024. Atlanta: Centers for Disease Control and Prevention (CDC); 2022.

9       Centers for Disease, C.; Prevention. Recovery of a patient from clinical rabies--Wisconsin, 2004. MMWR Morb Mortal Wkly Rep. 2004, 53, 1171-3.

10     Centers for Disease, C.; Prevention. Recovery of a patient from clinical rabies--California, 2011. MMWR Morb Mortal Wkly Rep. 2012, 61, 61-5.

11     Willoughby, R.E., Jr.; Tieves, K.S.; Hoffman, G.M.; Ghanayem, N.S.; Amlie-Lefond, C.M.; Schwabe, M.J.; Chusid, M.J.; Rupprecht, C.E. Survival after treatment of rabies with induction of coma. N Engl J Med. 2005, 352, 2508-14.

12     Burnett, E.J.; McPherson, R.J.; Aquin, J.P.; Xu, K.Y.; Plourde, P.J. Long-term serological response and boostability of intradermal rabies immunization: A retrospective chart review. Can J Public Health. 2024.

Reviewer 2 Report

Comments and Suggestions for Authors

This is overall a well written document that will add more information regarding the long-term persistence and kinetics of antibody response to human diploid cell rabies vaccine.  It adds useful information to the current literature, generally corroborating others' findings.  It appears to take advantage of vaccination efforts that had already been conducted for other purposes so this is a side benefit rather than the specific intent of those earlier vaccination goals.  

Some comments to help improve this paper for readers:

Not a critical comment, but in line 49 there is reference made to a better safety profile (favorable profile) of the four dose vaccine regimen relative to the five dose regimen with a reference regarding this statement.  No other discussion of any difference in safety profile was really described further in the paper, so if this is an important consideration of the manuscript, perhaps it would need to be further described.  

Line 89 - suggest word change from "one month time window." to "one month time interval from original vaccination." if that is the intended meaning.

Line 94 - suggest spelling out the acronym RVNA the first time it is used in the paper (Rabies Virus Neutralizing  Antibodies).

Table 1 - I must admit that it took me a while to comprehend the study design based on the multiple variables.  Two things might help the reader.  The first is putting the actual preliminary date for each of the groups in the next to last tier of boxes in the flowchart, e.g. 1-year after vaccination n=120, 2022).  Second is to put a reference to this figure in the methods section 2.2.

Comment mostly for the final copy editors - do not split Table 2 between two pages.

Figure 2, I am not sure what the meaning of "four stitches" and "five stitches" is based on standard English.  I checked to see if this was a colloquialism that I was not familiar with, but found none.  Suggest changing to "four shots" and "five shots"

Figure 2 - I am not sure why the legend below each whisker plot has the titles 1-3 years and 3+ years.  It seems confusing, since the reader is led to believe that the study groups are 1 year and 3 years. I suggest using these terms even if there is some small variation around this meaning.  IF there is in fact a different meaning, as in 1-3 years is truly that wide of a range, then there is a whole different discussion to be had.

Line 172 - suggest changing from "less participants" to "fewer participants" because the second term is a comparator.

Two general comments just for authors consideration and not critical to the paper.

First - one could speculate that there is a difference between the four vs. five regimen populations based on a general tendency to be "rules followers" in the 5 dose group vs. less stringency in the 4-dose group.  This speculation comes from two trending significant differences, the first being that more senior individuals and children are more likely to follow the requirement than the somewhat younger to middle aged 4 dose regimen group.  The second suggestive element is the level of influenza vaccine adherence between the two groups.  Again, just something that popped out to me.

Second - This question of how long rabies antibody persists in an individual made me wonder if there would be any scenario where a previously immunized individual, upon subsequent re-exposure to potential rabies virus, would not be advised to receive another round of vaccination?  Granted, there should be an anamnestic response based on prior immunity, but I suspect on a case-by-case basis, even previously immunized patients would be strongly recommended to obtain a booster.  Perhaps that should be mentioned in the article?  

Author Response

Reviewer #2
This is overall a well written document that will add more information regarding the long-term persistence and kinetics of antibody response to human diploid cell rabies vaccine.  It adds useful information to the current literature, generally corroborating others' findings.  It appears to take advantage of vaccination efforts that had already been conducted for other purposes so this is a side benefit rather than the specific intent of those earlier vaccination goals.  
Some comments to help improve this paper for readers:
Not a critical comment, but in line 49 there is reference made to a better safety profile (favorable profile) of the four dose vaccine regimen relative to the five dose regimen with a reference regarding this statement.  No other discussion of any difference in safety profile was really described further in the paper, so if this is an important consideration of the manuscript, perhaps it would need to be further described.  
    Response: We sincerely thank the Reviewer for their positive feedback and insightful comment. We acknowledge the importance of safety in the evaluation of the four-dose regimen. As noted in the manuscript, the study primarily focused on antibody persistence and kinetics, and because our research began after the completion of the vaccination, we did not have access to safety data. We agree that if safety is an important consideration, it should be further discussed. To address this, we have added the lack of safety data as a limitation in the manuscript. This addition highlights that while the four-dose regimen has been suggested to have a favorable safety profile, our study did not collect data on safety outcomes, as it relied on pre-existing vaccination efforts for PEP.

Revised Limitations Section:

The present study had several limitations. First, serum antibody titers before PEP vaccination could not be examined. Nevertheless, only rabies-vaccinated patients were included in this work, and a meta-analysis examining 80 studies found that essentially all immunologically naïve patients had pre-immunization serum RVNA concentrations below 0.5 IU/mL [1]. Furthermore, each individual only had one serum neutralizing antibody test at 1 or 3 years after vaccination, with a lack of data on short-term antibody titer and the dynamic change of serum RVNA over time. Finally, although previous studies have suggested that the four-dose Essen regimen has a favorable safety profile, safety data were not available in our study as we started after the completion of immunization, which limits our ability to comment on the safety aspects in this population.

Line 89 - suggest word change from "one month time window." to "one month time interval from original vaccination." if that is the intended meaning.
    Response: We sincerely thank the Reviewer for the helpful suggestion. We have revised the sentence to better reflect the intended meaning. The revised sentence now reads: Blood collection was performed at 1 and 3 years after PEP vaccination, with a one-month time interval from original vaccination.

Line 94 - suggest spelling out the acronym RVNA the first time it is used in the paper (Rabies Virus Neutralizing  Antibodies).
    Response: We sincerely thank the Reviewer for the suggestion. Upon review, we noted that the full-term rabies virus neutralizing antibodies (RVNA) is already spelled out in the introduction when the acronym is first introduced. We appreciate the Reviewer’s careful attention to detail and hope this clarifies the matter.

Table 1 - I must admit that it took me a while to comprehend the study design based on the multiple variables.  Two things might help the reader.  The first is putting the actual preliminary date for each of the groups in the next to last tier of boxes in the flowchart, e.g. 1-year after vaccination n=120, 2022).  Second is to put a reference to this figure in the methods section 2.2.
    Response: We sincerely thank the Reviewer for the constructive suggestions. To improve the clarity of the study design, we have made the following revisions: 1) We have included the actual preliminary date for each group in the flowchart, as suggested; 2) We have added a reference to the flowchart figure in section 2.2 of the Methods to ensure readers can easily locate the diagram and understand the study design.

Comment mostly for the final copy editors - do not split Table 2 between two pages.
    Response: We sincerely thank the Reviewer for the helpful comment. We understand that Table 2 should not be split across two pages, and we will ensure that this is addressed by the final copy editors during the preparation of the manuscript for publication.

Figure 2, I am not sure what the meaning of "four stitches" and "five stitches" is based on standard English.  I checked to see if this was a colloquialism that I was not familiar with, but found none.  Suggest changing to "four shots" and "five shots"
    Response: We sincerely thank the Reviewer for pointing this out. We have revised the terms "four stitches" and "five stitches" to "4-dose Essen group" and "5-dose Essen group," respectively, to maintain consistency across all tables and figures.

Figure 2 - I am not sure why the legend below each whisker plot has the titles 1-3 years and 3+ years.  It seems confusing, since the reader is led to believe that the study groups are 1 year and 3 years. I suggest using these terms even if there is some small variation around this meaning.  IF there is in fact a different meaning, as in 1-3 years is truly that wide of a range, then there is a whole different discussion to be had.
    Response: We sincerely thank the Reviewer for this valuable observation. We have revised the legend in Figure 2 to use the terms "1 year" and "3 years" as suggested, to avoid any confusion.

Line 172 - suggest changing from "less participants" to "fewer participants" because the second term is a comparator.
    Response: We sincerely thank the Reviewer for the helpful suggestion. We have revised the sentence to use "fewer participants" as recommended. The revised sentence now reads: In a study by Shankaraiah et al. [2], fewer participants completed the fifth dose of the Essen regimen than the fourth dose (60% vs. 69.8%).

Two general comments just for authors consideration and not critical to the paper.
First - one could speculate that there is a difference between the four vs. five regimen populations based on a general tendency to be "rules followers" in the 5 dose group vs. less stringency in the 4dose group.  This speculation comes from two trending significant differences, the first being that more senior individuals and children are more likely to follow the requirement than the somewhat younger to middle aged 4 dose regimen group.  The second suggestive element is the level of influenza vaccine adherence between the two groups.  Again, just something that popped out to me.
    Response: We sincerely thank the Reviewer for the thoughtful comments. We agree with the Reviewer’s observation that there may be underlying differences between the 4-dose and 5-dose groups, potentially driven by the tendency for individuals in the 5-dose group to be more "rule-following" compared to those in the 4-dose group. We also note the differences in age distribution and influenza vaccine adherence, which may introduce confounding biases in our study. Although these factors were not explicitly controlled for, we used multiple methods to adjust for potential confounders. To address this concern, we have added a discussion of these differences and the potential confounding bias to the discussion section, as well as a brief one to the limitations section of the manuscript.

Revised Discussion Section:

As an observational study, the choice of 4-dose or 5-dose Essen regimen was determined by participants, since we included participants who did not to receive the fifth dose irrespective of reason in the 4-dose group. Therefore, the disparity of baseline characteristics between the two groups somehow reflects the features of the patients. For instance, the patients completed 5-dose Essen regimen might have better adherence, with more attentive to their own health. It was supported by the higher proportion of participants with flu vaccination in the 5-dose Essen group than that in the 4-dose group (83.7% vs. 75.0%). Noticeably, the proportion of children and older adults was significantly higher in the 5-dose Essen group comparing to the 4-dose Essen group (32.6% vs. 21.7% and 34.8% vs. 18.9%, respectively), suggesting that age might be associated with adherence. Whether the patients' compliance and attention to their health can influence their medical behavior, and subsequently impact long-term prognosis, still requires further investigation.

Revised Limitations Section:

The present study had several limitations. First, serum antibody titers before PEP vaccination could not be examined. Nevertheless, only rabies-vaccinated patients were included in this work, and a meta-analysis examining 80 studies found that essentially all immunologically naïve patients had pre-immunization serum RVNA concentrations below 0.5 IU/mL [1]. Furthermore, each individual only had one serum neutralizing antibody test at 1 or 3 years after vaccination, with a lack of data on short-term antibody titer and the dynamic change of serum RVNA over time. Finally, although previous studies have suggested that the four-dose Essen regimen has a favorable safety profile, safety data were not available in our study as we started after the completion of immunization, which limits our ability to comment on the safety aspects in this population. Additionally, the information on sources of exposure was not collected and there were suggestive differences in the characteristics of the two groups (e.g., age distribution and influenza vaccination adherence) that may have introduced confounding biases. While these factors were adjusted for using various statistical methods, potential residual confounding cannot be ruled out.

Second - This question of how long rabies antibody persists in an individual made me wonder if there would be any scenario where a previously immunized individual, upon subsequent re-exposure to potential rabies virus, would not be advised to receive another round of vaccination?  Granted, there should be an anamnestic response based on prior immunity, but I suspect on a case-by-case basis, even previously immunized patients would be strongly recommended to obtain a booster.  Perhaps that should be mentioned in the article?  

Response: We sincerely thank the Reviewer for raising this interesting point. Indeed, the issue of whether a previously immunized individual requires a booster dose after re-exposure to potential rabies virus is important. According to the WHO guidelines, individuals who have received a complete rabies vaccination series (either pre- or post-exposure) typically require two booster doses (usually 1-site intramuscularly on day 0 and 3) if repeat exposure occurring >3 months after last vaccination. The rationale is that an anamnestic response will occur, rapidly boosting the immune response in these individuals. However, in some cases, particularly when there is uncertainty about the individual’s immune status (e.g., unknown vaccination history, significant time since last vaccination, or the initial immunization is incomplete), a complete immunization procedure after re-exposure may still be strongly recommended. This approach ensures that individuals are sufficiently protected in the event of re-exposure to the rabies virus. In the present study, no participant had subsequent exposure to rabies after initial immunization.

To address this, we have added a discussion of this topic in the Discussion section, highlighting the importance of considering individual factors, immune status, and the WHO guidelines when determining the need for a rabies booster dose after re-exposure.

Revised Discussion Section:

While both pre-exposure and post-exposure rabies vaccination is highly effective in inducing long-term immunity, re-vaccination following re-exposure is still needed in certain circumstances. According to the WHO guidelines, individuals who have received a complete rabies vaccination series (either pre- or post-exposure) typically require two booster doses (usually 1-site intramuscularly on day 0 and 3) if repeat exposure occurring >3 months after last vaccination [3]. This recommendation is based on the anamnestic immune response, where the immune system rapidly mounts a strong defense upon subsequent exposure. In a previous study, antibody titers after intradermal rabies vaccination declined over time, with seroprotection rates decreasing as much as 44% after 1.8-2.5 years. Interestingly, a booster dose after this period restores 100% seroprotection [4]. However, in certain situations (such as when the time since the last vaccination is prolonged, the individual's immune status is uncertain, or the initial immunization is incomplete), a complete immunization procedure after re-exposure may still be strongly recommended. This approach ensures that individuals are sufficiently protected in the event of re-exposure to the rabies virus. In the present study, no participant had subsequent exposure to rabies after initial immunization.

References

1       Xu, C.; Lau, C.L.; Clark, J.; Rafferty, A.C.; Mills, D.J.; Ramsey, L.; Gilbert, B.; Doi, S.A.R.; Furuya-Kanamori, L. Immunogenicity after pre- and post-exposure rabies vaccination: A systematic review and dose-response meta-analysis. Vaccine. 2021, 39, 1044-50.

2       Shankaraiah, R.H.; Rajashekar, R.A.; Veena, V.; Hanumanthaiah, A.N.D. Compliance to Anti-rabies Vaccination in Post-exposure Prophylaxis. Indian Journal of Public Health. 2015, 59.

3       World Health Organization. Rabies vaccines: WHO position paper, April 2018 - Recommendations. Vaccine. 2018, 36, 5500-3.

4       Burnett, E.J.; McPherson, R.J.; Aquin, J.P.; Xu, K.Y.; Plourde, P.J. Long-term serological response and boostability of intradermal rabies immunization: A retrospective chart review. Can J Public Health. 2024.

Reviewer 3 Report

Comments and Suggestions for Authors

This study investigates antibody persistence of human diploid cell rabies vaccine administered by the 4- versus 5-dose Essen intramuscular regimen in post-exposure prophylaxis in a prospective cohort study among Chinese population.

The study demonstrates that the 4- and 5-dose Essen regimens effectively produce durable immunogenicity, supporting the feasibility of implementing the 4-dose Essen regimen for rabies immunization in China.

Line 88: “Blood collection was performed at 1 and 3 years after initial vaccination”, I suggest replacing “after initial vaccination” with “after PEP vaccination”

I suggest that in Figure 1 the exclusion criteria from the study should be highlighted: there is no heading for that in the box.

Table 2: remove bolt character from the heading

Figure 2: legend, please replace “stitches” with injections or administrations

Line 201-203: please revise this sentence

Line 252: as in line 88, PEP should be mentioned

Author Response

Reviewer 3

This study investigates antibody persistence of human diploid cell rabies vaccine administered by the 4- versus 5-dose Essen intramuscular regimen in post-exposure prophylaxis in a prospective cohort study among Chinese population.
The study demonstrates that the 4- and 5-dose Essen regimens effectively produce durable immunogenicity, supporting the feasibility of implementing the 4-dose Essen regimen for rabies immunization in China.
Line 88: 
Blood collection was performed at 1 and 3 years after initial vaccination, I suggest replacing after initial vaccination with after PEP vaccination
    Response: We sincerely thank the Reviewer for the helpful suggestion. We have revised the sentence to better reflect the intended meaning. The revised sentence now reads: Blood collection was performed at 1 and 3 years after PEP vaccination, with a one-month time interval from original vaccination.

I suggest that in Figure 1 the exclusion criteria from the study should be highlighted: there is no heading for that in the box.
    Response: We sincerely thank the Reviewer for the suggestion. As recommended, we have added a heading for the exclusion criteria in Figure 1 to ensure clarity and improve the presentation of the study design.

Table 2: remove bolt character from the heading
    Response: We sincerely thank the Reviewer for the suggestion. As requested, we have removed the bold formatting from the heading in Table 2.

Figure 2: legend, please replace stitches with injections or administrations
    Response: We sincerely thank the Reviewer for pointing this out. We have revised the terms "four stitches" and "five stitches" to "4-dose Essen group" and "5-dose Essen group," respectively, to maintain consistency across all tables and figures.

Line 201-203: please revise this sentence
    Response: We sincerely thank the Reviewer for their comment. Upon reviewing the sentence, we believe the issue might be the phrasing and structure of the sentence. We have revised it to improve clarity and readability. Revised sentence: Overall, vaccination with HDCV, which demonstrates better immune persistence, using both the 5-dose and 4-dose Essen regimens, can provide durable immunogenicity. Additionally, the 4-dose Essen regimen offers a more convenient procedure.

Line 252: as in line 88, PEP should be mentioned

    Response: We sincerely thank the Reviewer for their comment. As suggested, we have included the mention of PEP in the revised sentence to ensure consistency and clarity throughout the manuscript. Revised sentence: First, serum antibody titers before PEP vaccination could not be examined.